biochemistry/analytical chemistry

ELISA, nanobody, SARS-CoV-2, Spike, site-specific conjugation

**Authors for correspondence:**
Raymond J. Owens
e-mail: ray.owens@strubi.ox.ac.uk
James H. Naismith
e-mail: james.naismith@strubi.ox.ac.uk

# The use of nanobodies in a sensitive ELISA test for SARS-CoV-2 Spike 1 protein

Georgina C. Girt[1,3], Abirami Lakshminarayanan[2,3], Jiandong Huo[1,2,3], Joshua Dormon[1], Chelsea Norman[1], Babak Afrough[4], Adam Harding[5], William James[5], Raymond J. Owens[1,2,3] and James H. Naismith[1,2,3]

[1]Structural Biology, The Rosalind Franklin Institute, Harwell Science and Innovation Campus, Didcot, UK
[2]Division of Structural Biology, University of Oxford, The Wellcome Centre for Human Genetics, Headington, Oxford, UK
[3]Protein Production UK, The Rosalind Franklin Institute – Diamond Light Source, The Research Complex at Harwell, Harwell Science and Innovation Campus, Didcot, UK
[4]National Infection Service, Public Health England, Porton Down, Salisbury, UK
[5]James and Lillian Martin Centre, Sir William Dunn School of Pathology, University of Oxford, Oxford, UK

GCG, 0000-0001-9936-1129; AL, 0000-0002-1688-3667;
JHN, 0000-0001-6744-5061

Detection of severe acute respiratory syndrome coronavirus 2 (SARS-CoV-2) antigens in the fluid has important uses in biotechnology, and is integral to many point-of-care SARS-CoV-2 diagnostics. Sandwich enzyme-linked immunosorbent assays (ELISAs) are a sensitive, well-established method of measuring antigens in solutions. They use one ligand to capture and the other ligand to detect the target analyte. Detection is commonly achieved using colorimetric readout obtained upon the reaction of a substrate with HRP-conjugated secondary ligand. Nanobodies, the $V_HH$ domain of camelid antibodies, have expanded the repertoire of molecules used in antigen detection. Nanobodies' high affinity for target antigens, their compact structure, their high stability and ease of production has driven research into their use as diagnostic reagents. Guided by a structural understanding of epitopes on the receptor-binding domain of the SARS-CoV-2 Spike protein, we investigated various combinations of engineered nanobodies in a sandwich ELISA to detect the Spike protein of SARS-CoV-2. We have identified an optimal combination of nanobodies. These were selectively functionalized to further improve antigen capture, enabling the measurement of sub-picomolar amounts of SARS-CoV-2 Spike protein in solution. With this combination, the routine detection limit in samples inactivated by heat and detergent corresponded to less than seven focus-forming units of infectious SARS-CoV-2.

# 1. Background

Severe acute respiratory syndrome coronavirus 2 (SARS-CoV-2) is the novel human coronavirus (HCoV) responsible for the COVID-19 pandemic. SARS-CoV-2 is a positive sense, single-stranded (RNA) virus with a 30-kb genome, encoding four key structural proteins: the nucleocapsid (N) protein that holds the RNA genome, as well as E (envelope), M (membrane) and S (Spike) proteins. S protein is expressed on the surface of the virus particle and through its receptor-binding domain (RBD) binds to angiotensin-converting enzyme 2 (ACE2) receptors [1]. People infected with the virus display symptoms ranging from essentially none (asymptomatic) to severe disease leading to mortality.

The viral Spike protein is at the centre of much of the biotechnology being related to SARS-CoV-2. Spike protein, generated in humans by the introduction of genetic material encoding it, is the basis of vaccines, both currently licensed [2,3] and in development [4]. Direct injection of the Spike protein itself (attached to nanoparticles) has shown promising results as a vaccine [5], and highly pure Spike protein is used to monitor for the presence of antibodies against the protein [6], thereby identifying previously infected or vaccinated individuals. A rapid quantification method specific for well folded 'high quality' SARS-CoV-2 Spike protein, therefore, has many applications.

Containing the spread of the virus relies on diagnostic tests particularly in identifying asymptomatic infection. There are two main diagnostic approaches for SARS-CoV-2. The first and still considered the most accurate approach detects viral RNA by PCR amplification using DNA primers and reverse transcription [7]. RT-PCR and qPCR are considered the gold standard by the FDA and ECDC due to their superior sensitivity and specificity. Improvements in sensitivity and simplicity have come from loop-mediated isothermal amplification (LAMP) [8]. The second method is to directly detect viral components (antigens) most commonly a specific viral protein not found in uninfected humans. Antigen tests are commonly found in point-of-care devices used to diagnose several respiratory pathogens, including bacterial (streptoccocal), viral (influenza, HIV) and parasitic (malaria) infections [9]. Such point-of-care assay devices are now in widespread use for the detection of SARS-CoV-2 infection [10]. Antigen tests in these devices often originate from laboratory-based sandwich enzyme-linked immunosorbent assays (ELISA). These systems employ a capture agent and a detection agent; the requirement is that both bind to non-overlapping regions of the antigen. Antibodies are most commonly used in ELISA tests due to their high sensitivity, specificity, established manufacturing processes and safety. The sensitivity of ELISA is dependent on factors including the binding affinity between antigen and the antibodies, the selectivity of the antibodies (no non-specific interactions or off-target binding), and whether antigen-capture leads to occlusion of the other epitope and the colour change system. In point-of-care devices (also known as lateral flow immunoassays (LFIs)), the capture antibodies are often immobilized on nitrocellulose and the probe antibody, modified with a colorimetric nanoparticle, is incubated with the test sample, thus becoming the 'mobile phase' of the device. Thus, in addition to the constraints of ELISA, there are important technological and manufacturing factors that need attention to develop a diagnostic device [11]. However, where lateral flow devices are intended to give only a positive or negative result, ELISA-based systems are inherently quantitative and have a wide dynamic range. There are currently 36 approved lateral flow devices that have passed regulatory evaluation in the UK.

Nanobodies (Nbs, also known as the $V_H$Hs) are the variable antigen-binding region that arise in single-domain antibodies (sdAbs) derived from a single heavy chain antibody variant found in camelids and in cartilaginous fish. The small size (13–15 kDa) and stability of nanobodies makes them an attractive alternative to antibodies. Nanobodies have revolutionized structural biology, and have entered clinical trials as therapies and tools [12]. We have previously reported nanobodies derived by DNA shuffling affinity maturation from a parent nanobody, identified from a naive llama $V_H$H library that bind to the Spike protein [13]. One of these, H4, was shown to neutralize SARS-CoV-2 with nM potency [13]. We have recently reported another set of nanobodies with pM binding affinity (agents C1, F2, C5) [14]. Structural analysis of nanobodies revealed that one group (cluster 2 antibodies [15]) (H4, C5) bound to an epitope on the RBD that overlapped with the ACE2 binding epitope [13,14]. A second group (cluster 1 antibodies [15]) (C1, F2) bound to a non-overlapping epitope at a different location on RBD [14]. This second site was first disclosed by the potent neutralizing human antibody CR3022 [16,17]. We have previously shown when CR3022 and H4 were combined they gave additive neutralization [13].

To account for limitations of nanobodies in ELISA, such as poor adhesion to the plate via passive adsorption and obstruction of the binding motif due to their small size, we prepared $V_H$H conjugates with human IgG1 Fc (bivalent and glycosylated). This not only increased the overall size of the

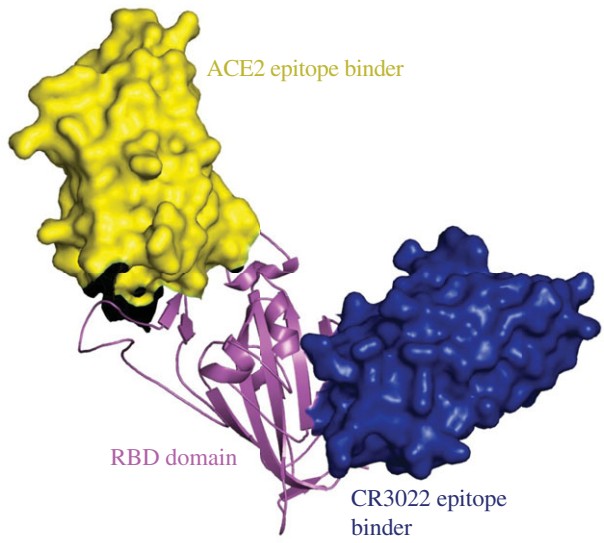

**Figure 1.** The two non-overlapping epitopes used for the ELISA. In yellow space fill is a nanobody that recognizes the ACE2 RBD epitope [13,14] known as cluster 2 [15] (group 1 [18]) epitope and in blue a second nanobody [14] which recognizes a different RBD epitope, first described in the CR3022 structure [16,17], known as cluster 1 [15] (group 4 [18]) epitope.

nanobodies (while still keeping them significantly smaller than antibodies), but also introduced regions that could be site selectively modified distal to the $V_HH$ domain, encouraging adhesion to the plate in optimized orientation, away from the antigen-binding motif.

Guided by our structural insights, we tested pairs of agents (one at each epitope) for their use in an ELISA (figure 1). We identified a desirable preparation for reagents and conditions for attachment to ELISA plates. The optimally performing pair were tested against samples of combined heat and detergent-inactivated SARS-CoV-2 virus, a chemically inactivated virus sample, an X-ray inactivated sample and live, replication-incompetent pseudotyped virus.

# 2. Results

## 2.1. Optimizing the capture component

We first investigated the detection of purified Spike protein in solution using the pM binder C1-Fc (cluster 1 ACE2 epitope binder, $V_HH$ conjugated to human IgG1 Fc (bivalent and glycosylated)) and nM binder H4 (cluster 2 CR3022 epitope binder, $V_HH$ domain only) [13] in our ELISA assay, with ABTS (2,2′-Azinobis [3-ethylbenzothiazoline-6-sulfonic acid]-diammonium salt) as horse radish peroxidase (HRP) substrate. Using C1-Fc as the capture agent, with H4-HRP as the detection probe gave a limit of detection (LOD) of $1.85 \times 10^6$ pg ml$^{-1}$ (1.85 µg ml$^{-1}$) of Spike protein (estimated greater than 3.3 times the standard deviation of blank samples). When H4 was adsorbed onto the plate and the HRP-conjugates of C1-Fc were used as the probe, the detection limit was $20 \times 10^6$ pg ml$^{-1}$ (20 µg ml$^{-1}$) of Spike protein (figure 2a).

We then investigated whether biotinylated nanobodies binding to streptavidin-treated plates improved the sensitivity of the assay. High binding Nunc plates were treated with unmodified C1-Fc and separately streptavidin-coated plates were treated with biotinylated C1-Fc (biotin$_x$-C1-Fc) prepared with a Sulfo-NHS-Biotin reagent. This reagent biotinylates lysine residue side chains in a non-specific stochastic manner. Both plates were treated with recombinant Spike glycoprotein ranging from 100 µg ml$^{-1}$ to 0.25 µg ml$^{-1}$, and then probed with H4-HRP. Where C1-Fc was passively absorbed the LOD of S protein was calculated as $1.3 \times 10^6$ pg ml$^{-1}$, essentially the same result as a standard plate. Using the biotin$_x$-C1-Fc with streptavidin plates lowered the LOD to $100 \times 10^3$ pg ml$^{-1}$ (figure 2b).

## 2.2. Evaluating nanobody pairs

Having established better antigen detection using biotinylated nanobodies, we continued with streptavidin-coated plates using the capture nanobody in the form biotin$_x$-Nb-Fc (0.5 µg ml$^{-1}$) and the

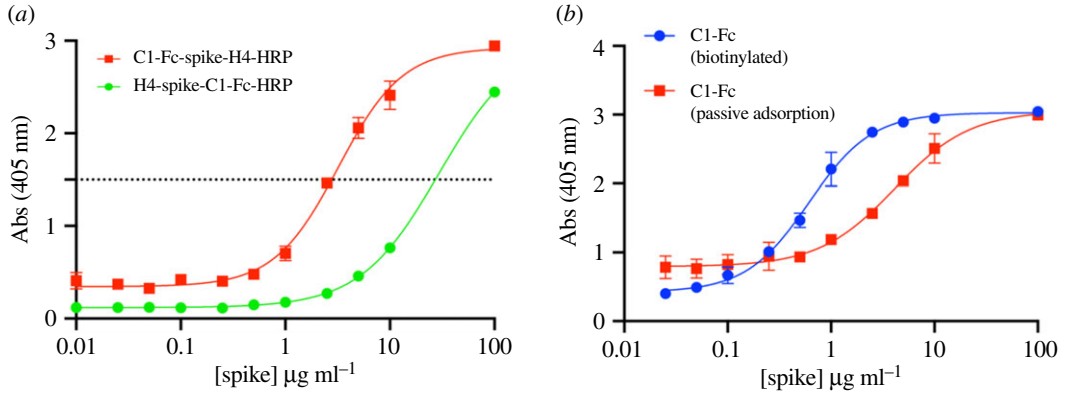

**Figure 2.** Immobilizing the capture agent by passive absorption versus biotinylation. (*a*) Using H4 passively absorbed onto the plate as capture agent and C1-Fc as the probe is shown in green. Reversing capture and probe antibodies is shown in red. (*b*) C1-Fc as capture and H4-HRP as the probe but using different ELISA plates. In red is the passive absorption of C1-Fc. Shown in blue is using biotin$_x$-C1-Fc bound to streptavidin plates. Absorbance values above the dotted line were not actually measured, rather the sample was first diluted then measured. The signal was produced by the reaction of HRP with ABTS.

**Table 1.** The calculated limit of detection of Spike protein in pg ml$^{-1}$. In parenthesis is the slope of the line. In italics is the combination selected.

| | probe | | |
|---|---|---|---|
| capture | C1-Fc-HRP | F2-Fc-HRP | C5-Fc-HRP |
| biotin$_x$-C1-Fc | —— | 9000 (12) | 100 (64) |
| biotin$_x$-F2-Fc | 1000 (9) | —— | 400 (79) |
| biotin$_x$-C5-Fc | 200 (47) | *500 (79)* | —— |
| biotin$_x$-H4-Fc | 5000 (2) | 3000 (8) | 3000 (6.5) |

probe nanobody HRP-Nb (0.5 µg ml$^{-1}$). We introduced a further two nanobodies: F2 (cluster 1 ACE2 epitope binder) and C5 (cluster 2 CR3022 epitope binder). We switched to use 3,3′,5,5′-tetramethylbenzidine (TMB) as the substrate of HRP, since this commercially available molecule has higher sensitivity and faster colour development than ABTS [19]. We found that H4-Fc-HRP when used as the probe gave unreliable detection results. Dot blotting of H4-Fc-HRP with other nanobodies suggested it may have non-specific interactions; the other nanobodies did not show this behaviour. The results from the various remaining combinations are given in table 1 and figure 3. The estimated LOD varied from $5 \times 10^3$ pg ml$^{-1}$ (biotin$_x$-H4-Fc, C1-Fc-HRP) down to 200 pg m$^{-1}$ (biotin$_x$-C5-Fc, C1-Fc-HRP) depending on the combination (table 1). As LOD is dependent on the standard deviation (s.d.) of replicates, the slope of linear regression was used in place, with a higher regression value indicating greater sensitivity. The biotin$_x$-C5-Fc, F2-Fc-HRP gave a regression gradient (79)—comparable to the reverse pairing biotin$_x$-F2-Fc, C5-Fc-HRP—but since the former combination had higher overall absorbance, it was selected as the optimal combination.

As a negative control, we repeated the ELISA using nanobody pairings with overlapping epitope combinations. These were as expected much less sensitive and when plotted gave lines with significantly less steep slopes, thus they did not respond as strongly to changes in concentration.

## 2.3. Testing against virus

Having tested nanobodies against purified recombinant Spike protein, we went on to test Spike antigen displayed on the viral surface using our best capture–detection nanobody pairs, namely C5-Fc with F2-Fc-HRP in the ELISA assay. Viral titres were quantified by either tissue culture infectious dose 50 (TCID50) assay, or focal forming assay (FFA). We were unable to test live virus due to safety

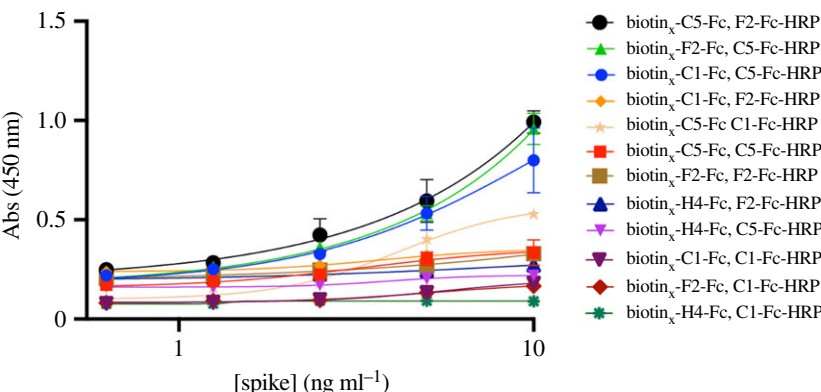

**Figure 3.** Pairings of biotinylated nanobodies as capture (biotin$_x$-XX-Fc) and probe (YY-Fc-HRP) with Spike protein. The signal was produced by the reaction of HRP with TMB.

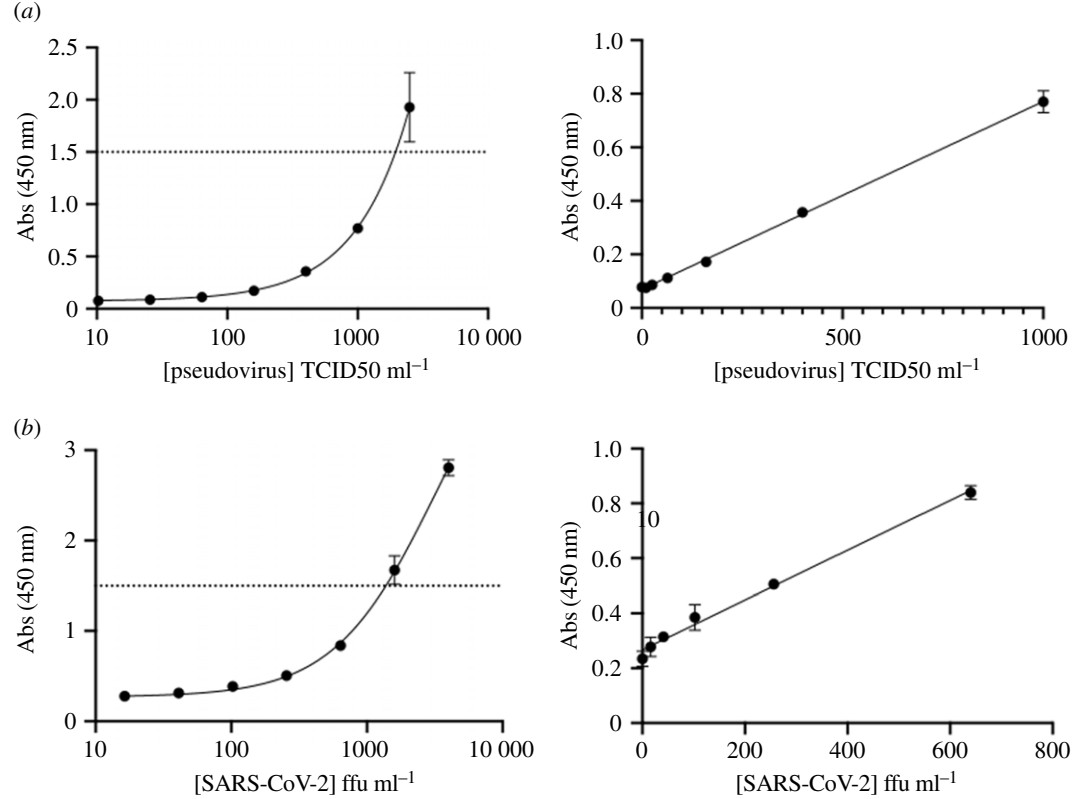

**Figure 4.** ELISA using biotin$_x$-C5-Fc and F2-Fc-HRP combination we were able to measure antigen when presented in the viral form. (a) Pseudotyped virus was detected to 21 TCID50 ml$^{-1}$. (b) Heat-Empigen inactivated virus was detected at 103 ffu ml$^{-1}$. Absorbance values above the dotted line were not actually measured, rather the sample was first diluted then measured.

restrictions. Instead, we tested intact pseudotyped NL4.3 HIV-1 backbone virus displaying surface SARS-CoV-2 S glycoproteins. This pseudovirus is based on the genomic backbone of HIV-1, with two frameshifts which render it Env (viral envelope protein) negative and Vpr (viral protein R) negative. Without the expression of these two proteins, this clone is competent for a single round of replication and non-infectious. Our lower LOD was 21 TCID50 ml$^{-1}$ of infectious pseudotyped virus (figure 4a, table 2). We were able to test inactivated WT SARS-CoV-2 virus; three different methods of inactivation were chosen: (i) 4% formaldehyde (FA) (ii) X-ray irradiation (12.19 kGy) (iii) Empigen (a zwitterionic detergent) (0.05%) and heat (60°C, 30 min). We were unable to detect FA inactivated SARS-CoV-2 but detected heat-Empigen inactivated virus at 103 ffu ml$^{-1}$ (figure 4b, table 2). The ELISA was not as sensitive for the X-ray irradiated samples, which were detected above 305 pfu ml$^{-1}$ (electronic supplementary material, figure S1) table 2.

**Table 2.** The calculated limit of detection for pseudovirus and heat-Empigen inactivated SARS-CoV-2.

| | slope | s.d. | LOD | $r^2$ |
|---|---|---|---|---|
| pseudovirus | 0.0007019 | 0.00442 | 21 TCID50 ml$^{-1}$ | 0.996 |
| SARS-CoV-2 (heat-Empigen) | 0.000890 | 0.0277 | 103 ffu ml$^{-1}$ | 0.980 |

## 2.4. ELISA with site-selectively biotinylated nanobodies

We evaluated whether site-specific regioselectively controlled biotinylation of the capture agents would improve the sensitivity of the assay by ensuring consistent orientation of the nanobody binding site outward into the well and avoiding any lysine biotinylation proximal to the epitope binding site which could potentially hinder the nanobody–Spike interaction [20].

Site-specific functionalization of antibodies by tagging native antibody residues has been achieved using microbial transglutaminase (mTGase) [20,21]. mTGase recognizes and tags the glutamine residue in the conserved sequence 'PREEQYNXT' in the Fc region of antibodies. Previous studies have used this method for site-specific conjugation of radioactive probes, fluorescent dyes as well as introducing orthogonal functional tags in the Fc region of antibodies [22]. Since the capture nanobody, namely C5Fc, used in our ELISA is fused with an Fc tail and hence has the conserved 'tag' sequences, we decided to employ enzymatic transglutamination to specifically functionalize the glutamine residue of the PREE**Q**YNST sequence in the Fc region (figure 5a). In order to better expose the target glutamine residue for reaction, we cleaved the N-linked glycans using PNGase F digestion. Amine-PEG3-Biotin was added to the PNGase digested antibodies along with the transglutaminase enzyme and incubated at 37°C. The progress of the reaction was monitored by mass spectrometry (figure 5b and electronic supplementary material, figure S6). In addition to the required product, we consistently observed another product in our reactions which showed a loss of mass of 15 Da. We believe that the addition of an internal lysine residue was a competing reaction which resulted in the observed product with a loss of mass when compared to the deglycosylated C5-Fc.

A streptavidin-coated 96-well plate was treated with biotin$_x$-C5-Fc and site-specific biotinylated C5-Fc (C5-Fc-SS-biotin) at the same concentration. We used recombinant SARS-CoV-2 S glycoprotein (figure 6a and electronic supplementary material, figure S3a), RBD (figure 6b and electronic supplementary material, figure S3b), pseudovirus (figure 6c and electronic supplementary material, figure S3c) and heat-Empigen inactivated WT SARS-CoV-2 (figure 6d and electronic supplementary material, figure S3d) as substrates. The plates were probed with F2-Fc-HRP. Using C5-Fc-SS-biotin as the capture showed an increased sensitivity compared to the multi-biotinylated form for purified Spike protein (147 pg ml$^{-1}$ versus 514 pg ml$^{-1}$) (table 3, electronic supplementary material, figure S2a), and for purified RBD (33 pg ml$^{-1}$ versus 85 pg ml$^{-1}$) (table 3, electronic supplementary material, figure S2b), consistent with the lower molecular weight of the isolated domain. Only a minor increase in sensitivity was observed in for pseudovirus (16 TCID50 ml$^{-1}$ versus 21 TCID50 ml$^{-1}$) (table 3, electronic supplementary material, figure S2c). With the same batch of heat-Empigen inactivated SARS-CoV-2 sensitivity improved from 103 to 16 ffu ml$^{-1}$ (table 3, electronic supplementary material, figure S2d). In order to test the reproducibility of the assay, we prepared multiple independent batches of heat-Empigen inactivated SARS-CoV-2 and C5-Fc-SS-biotin, the LOD was observed to vary by up to 69 ffu ml$^{-1}$ (electronic supplementary material, figure S4). We attribute the variability arising from the preparation of the virus (table 3).

## 3. Discussion

In recent years, nanobodies have gained increased interest as replacements for antibodies due to their smaller size, highly modular nature, enhanced stability and relative ease of production [12,23]. We sought to determine if we could apply a classical double-sandwich ELISA format to nanobodies against the Spike protein of SARS-CoV-2. An accurate ELISA could be useful in the biotechnology industry for quantifying Spike protein during its production.

Direct absorbance of the $V_H H$ domain of the nanobody onto plates was not successful in an ELISA against the Spike protein. We concluded either $V_H H$ was not readily adsorbed onto the plate or when absorbed onto the plate its binding site was obscured. Other workers [24] targeting a different protein have observed similar issues with $V_H H$ domains and drawn similar conclusions. Switching to biotinylated protein and streptavidin-coated plates gave much better results. The use of Fc fusion

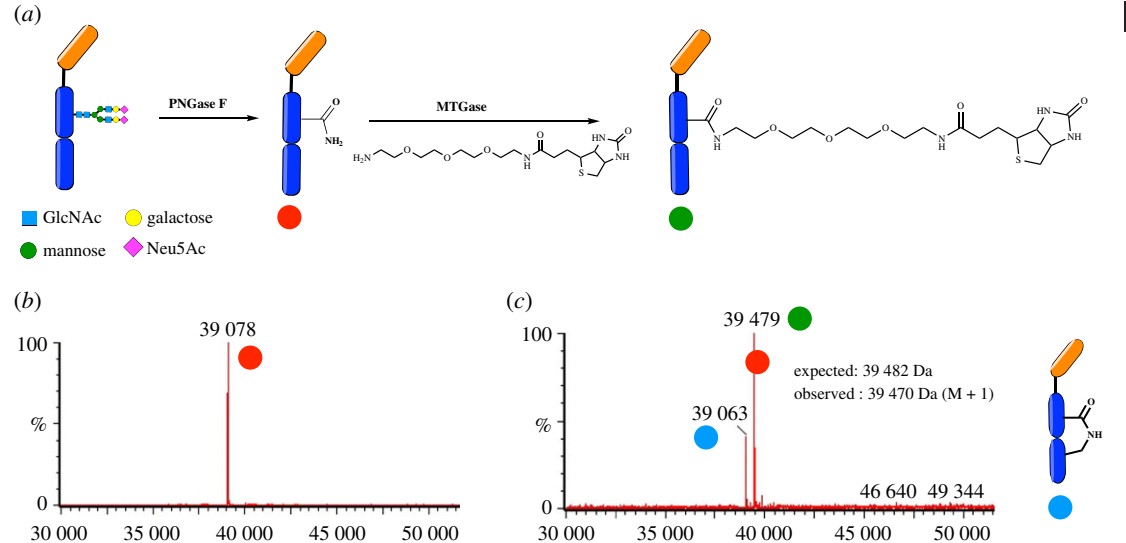

**Figure 5.** Site-specific functionalization of C5Fc with PEG-biotin using mTGase mediated transglycosylation. (*a*) Schematic of the reaction. (*b*) The C5Fc nanobody was deglycosylated using PNGaseF enzyme to expose the target 'Q' residue and deglycosylation was monitored by mass spectrometry. (*c*) Trannsglutamination of C5Fc upon the reaction of amine-PEG3-biotin with the deglycosylated nanobody using microbial transglutaminase (mTGase). The addition of internal lysine residue was a completing reaction.

simplifies the biotinylation strategy since the Fc portion has multiple lysine residues, enabling rapid amine functionalization using standard protocols. Furthermore, the Fc region has native conserved residues which can be tagged enzymatically to afford selective biotinylation with advantages of better surface orientation. We identified the optimal combination for detection of Spike protein to consist of biotin$_x$-C5-Fc as the capture agent with F2-Fc-HRP as the probe agent. This combination gave an LOD of 514 pg ml$^{-1}$ (table 1). The ELISA showed a linear response indicating it would be suitable for reliably quantitating Spike. Other nanobody combinations also gave a LOD below 1000 pg ml$^{-1}$, demonstrating their high specificity. By using site-selective biotinylation the LOD was reduced to 147 pg ml$^{-1}$ for Spike protein. The use of the nanobody pairs thus gave an ELISA that is simple to use as a laboratory tool to monitor the heterologous production of Spike protein. We have shown that C5 has a potent therapeutic effect when administered topically or intraperitoneally to hamsters [14]. This, combined with our previous observation of additive neutralization between non-overlapping neutralizing agents [13], suggests that C1 or F2 could be effective in combination with C5.

As new variants of the Spike protein become important, for example carrying the E484 K mutant [25], the probe nanobody, C5, will require modulation to accommodate newer antigenic epitopes. We have shown elsewhere that C5 does not bind to protein with the E484 K [14], although it does bind to the N501Y [26] mutant [14]. The C1 nanobody was shown to bind equally well to multiple variants of the virus [14]. Repeating the assay with the beta and delta variant RBD (electronic supplementary material, figure S7) shows as expected that the delta variant RBD is detected but the beta variant RBD is not. The advantage of nanobodies is that they are relatively straightforward to raise (either by llama immunization [14] or phage display approaches [13]) against new antigens in comparison to their human antibodies. Nanobody technology thus offers a robust and adaptable assay platform for process monitoring.

We verified that the ELISA was also compatible with the Spike protein when presented as an intact viral particle by using pseudotype virus confirming that the nanobody can recognize natively folded protein as part of an infectious virus. We also obtained evidence that the assay was sensitive to the 'quality' of protein, virus inactivated by FA (which is expected to chemically alter the binding epitopes) was not detected. SARS-CoV-2 virus inactivated with the mild detergent Empigen [24] was detected with a sensitivity of 103 ffu ml$^{-1}$ (140 TCID50 ml$^{-1}$). We excluded the possibility that Empigen was responsible for the gain by showing it had no effect on the detection of Spike protein (electronic supplementary material, figure S3). Using the reported estimate of 1000 virons in an ffu [27], 25 spikes in each virion [28] and 600 kDa as the weight of the Spike trimer, this corresponds to around 30 pg of Spike per ml. We attribute this 20-fold gain the polyvalent presentation of the Spike protein on the viral membrane and thus we gain from avidity. Pseudovirus, where the protein is presented on the viral membrane, was detected at 21 TCID50 ml$^{-1}$, the very low threshold consistent with an avidity driving a gain in the sensitivity of the ELISA.

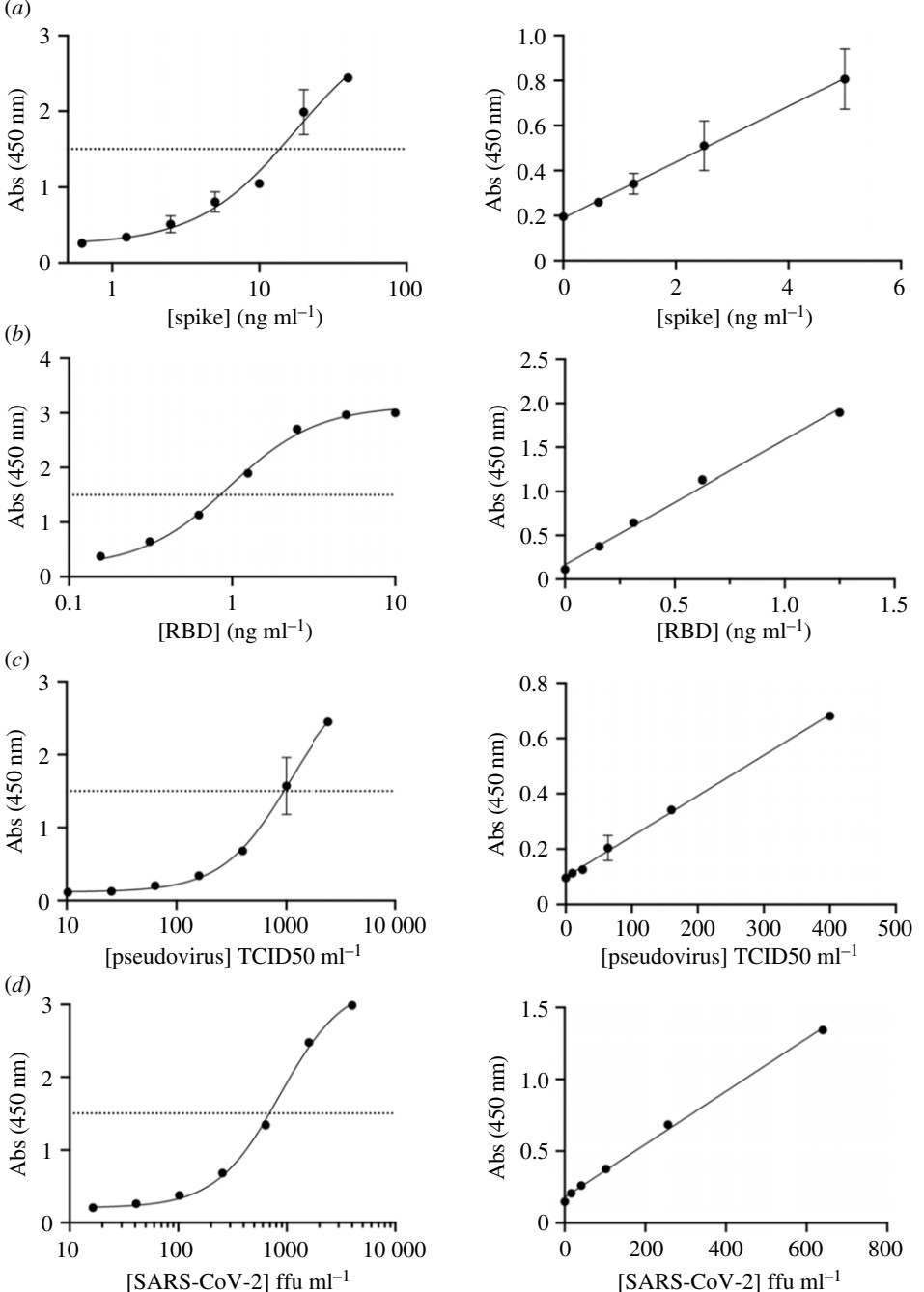

**Figure 6.** ELISA using C5-Fc-SS-biotin and F2-Fc-HRP against two recombinant antigens and two viral samples (*a*) Spike protein was detected to 147 pg ml$^{-1}$. (*b*) RBD was detected to 33 pg ml$^{-1}$ (*c*) Pseudotyped virus was detected to 16 TCID50 ml$^{-1}$. (*d*) Heat-Empigen inactivated virus was detected to 16 ffu ml$^{-1}$, see also electronic supplementary material, figure S4. Absorbance values above the dotted line were not actually measured, rather the sample was first diluted then measured.

**Table 3.** The calculated limit of detection for Spike protein, RBD, pseudovirus and heat-Empigen inactivated SARS-CoV-2 using C5-Fc-SS-biotin. See also electronic supplementary material, figure S4.

|  | slope | s.d. | LOD | $r^2$ |
|---|---|---|---|---|
| Spike | 0.1241 | 0.00552 | 147 pg ml$^{-1}$ | 0.911 |
| RBD | 1.421 | 0.0143 | 33 pg ml$^{-1}$ | 0.993 |
| pseudovirus | 0.001466 | 0.00701 | 16 TCID50 ml$^{-1}$ | 0.993 |
| SARS-CoV-2 (heat-Empigen) | 0.001843 | 0.00889 | 16 ffu ml$^{-1}$ | 0.997 |

In order to probe the robustness of the assay, we used four independent batches of heat-Empigen inactivated SARS-CoV-2 with our site-selective C5-Fc-SS-biotin. These viral batches varied by the time harvested post-infection (72 h or 48 h) and host cells for propagations (Vero CCL81 or Vero E6-TMPRSS2). We observed a weak correlation between incubation time and ELISA sensitivity, with longer incubations tending to lower limits of detection. Since purified RBD and Spike proteins give highly consistent results, we attribute the observed assay variability to viral batches. The observed LOD ranged from 16 ffu ml$^{-1}$ to 69 ffu ml$^{-1}$ (electronic supplementary material, figure S4). In absolute terms, the C5-Fc-SS-biotin and F2-Fc-HRP combination was able to detect less than 7 ffu of heat-Empigen inactivated virus.

# 4. Conclusion

Detecting SARS-CoV-2 antigens has played a major role in containing the pandemic by identifying infected individuals through mass testing. The FDA and EUA approved rapid detection instruments with the lowest reported limits of detection are Abbott Diagnostics BinaxNOW COVID-19 Ag Card and LumiraDx UK SARS-CoV-2 Ag Test (22.5 TCID50 ml$^{-1}$ and 32 TCID50 ml$^{-1}$, respectively). A sandwich ELISA has important differences to the lateral flow devices and translation into a clinically useful device is not simple. However, the sensitive detection of virus we have observed with commercially available HRP reagents and simple ELISA plates demonstrates the potential utility of nanobodies as diagnostics. There is scope to further improve the assay: we used TMB as the HRP substrate but there are alternatives to HRP conjugation that could be deployed and are reported to enhance the sensitivity of the technique, for example, metal oxides [29]. Nanobodies themselves have been engineered in other multivalent forms, including trimers [30]. Our data showing the enhancement of sensitivity with heat-Empigen inactivated virus suggests avidity of analyte is important, thus higher-order multimers capture/detection agents may give further gains.

# 5. Material and methods

## 5.1. Chemicals

PNGase F was purchased from NEB and mTGase was sourced from Zedira (T001). Amino-PEG3-biotin was purchased from ThermoFisher. MES buffer and SDS PAGE gels were purchased from Invitrogen. Vivaspin filter membranes were purchased from Sartorius and the membranes were washed with milliQ water and phosphate-buffered saline (PBS) before use. Other chemicals were purchased from SigmaAldrich unless otherwise stated.

## 5.2. Protein production

Purified H4, H4-Fc, C5-Fc, C1-Fc, F2-Fc, RBD and SARS-CoV-2 Spike were prepared as previously described [13,14]; these reports also contain the nanobodies sequences. HRP-nanobody conjugates were prepared according to the method supplied with Abcam Lightning Link HRP conjugation kit (www.abcam.com) and used without any further treatment. The extent of conjugation was estimated by analysing the bands of conjugated nanobody in SDS–PAGE gel electrophoresis. The addition of biotin non-specifically to lysine residues of the nanobodies was achieved using ThermoFisher EZ-Link Sulfo-NHS-Biotinylation Kit or EZ-Link Sulfo-NHS-LC-Biotinylation Kit (www.thermo.com). The kits were used according to the manufacturers protocol. Once complete, the reaction solutions were dialyzed in PBS to remove any unreacted biotin reagent, and concentration determined by nanodrop. The extent of biotinylation was established with Thermo Scientific Fluorescence Biotin Quantification Kit; this was typically four to six biotin moieties for Fc-conjugated nanobodies, and one to three biotin moieties for VHH domain nanobodies.

## 5.3. Site-specific labelling

N-linked glycans on 500 µg C5-Fc (1 mg ml$^{-1}$ in PBS) were removed by incubation at 37°C for 18 h with 5 µl of PNGaseF enzyme in PBS (electronic supplementary material, figure S5a) [20,21]. The completion of the reaction was determined by gel electrophoresis (electronic supplementary material, figure S5b). Deglycosylated C5-Fc (dgC5-Fc) was purified on a protein A affinity column (GE lifesciences) and

eluted with citrate buffer (pH 3) and neutralized using 1 M Tris (pH 9) (electronic supplementary material, figure S5c). The purified dgC5-Fc was buffer exchanged into PBS using a 10 kDa Vivaspin filter membrane. One-hundred micrograms of dgC5-Fc (1 mg ml$^{-1}$) was then incubated at 37°C with amino-PEG3-Biotin (100 equivalents) and mTGase enzyme (6 U ml$^{-1}$) (modifies residue Q in the sequence PREE**Q**YNST) for 4 h. one microlitre aliquots of the reaction were collected and analysed by reducing LCMS (concentration of 0.002 mg ml$^{-1}$ protein in 20 mM DTT). The modified nanobody was analysed on a ProSwift RP-4H HPLC column using 0.1% aqueous formic acid and 95% acetonitrile as mobile phases. The data were deconvoluted and analysed using MassLynx v. 4 software.

## 5.4. Virus samples

### 5.4.1. SARS-CoV-2 WT (heat-Empigen and 4%FA) and pseudovirus

Infectious SARS-CoV-2 virus was grown in Vero CCL81 or Vero E6-TMPRSS2. Propagation was performed in T175 tissue culture flasks. When cells reached approximately 70% confluence, all medium was removed and 1 ml (DMEM, 1% FBS, 1% P/S) of the virus-containing medium was added to each flask (MOI ∼ 0.001). This was left to incubate for 10 min at room temperature before topping up with a further 19 ml of medium.

Flasks were left to incubate for 48–72 h post-infection until significant cytopathic effect was visible in the culture. Harvesting consisted of pooling the media, pelleting cell debris (5 min, 500 RCF) and subsequent aliquoting of the supernatant.

Virus titre was determined by focus-forming assay. Vero CCL81 cell suspension was added to serially diluted virus stock and incubated for 2 h. A viscous carboxymethylcellulose (CMC) overlay was added and plates incubated a further 22 h. The medium was then removed, monolayers fixed with 4% FA (30 min), and cells stained for SARS-CoV-2 nucleocapsid by a standard primary/secondary-HRP procedure. Titre was measured as focus-forming units per ml (ffu ml$^{-1}$) [31]. Pseudovirus was produced by co-transfection of HEK-293T cells with a pNL4–3 (ΔEnv, luciferase) lentiviral plasmid and a plasmid encoding the SARS-CoV-2 Spike gene. Transfection was with polyethylenimine (PEI) for 4 h. After which, cells were washed, fresh media (DMEM, 10% FBS, 1% P/S) applied and virus production left to proceed for 48 h. The virus was harvested by pooling supernatant and concentration by polyethylene glycol (PEG) precipitation. Titration of pseudovirus was performed by infecting cells overexpressing ACE2 (MDCK-ACE2) for 48 h with a limiting dilution of virus stock. The readout was by standard luciferase assay of cell lysate, with each well classified as positive or negative for infection. Titre was measured in units of TCID50 ml$^{-1}$. Inactivation of infectious virus stocks was carried out either by FA fixation or combined heat/detergent inactivation. For FA fixation: 32% stock solution was mixed with virus sample to achieve a final 4% FA concentration and incubated for 30 min before removal from the CL3 facility. For heat/detergent inactivation: Empigen detergent was diluted in MilliQ water to a 5% solution. This was mixed 1 : 10 with the virus sample (final Empigen concentration 0.5%). The sample was then heated for 30 min at 56°C.

### 5.4.2. SARS-CoV-2 WT (X-ray inactivated)

X-ray inactivated SARS-CoV-2, isolate England/02/2020 (passage 3) cultured in Vero h/SLAM (ECACC accession: 04091501) at a pre-irradiation titre of $1 \times 10^6$ pfu ml$^{-1}$ was provided by Public Health England, Porton Down. Briefly, the virus was irradiated with 11 kGy of X-rays at +4°C using a MultiRad 225 KeV system (Precision X-rays, USA). Absorbed dose was collected in real time using a UNIDOS E reference dosimeter fitted with a Semiflex 31010 ionization chamber. Inactivation of the virus was demonstrated by a reduction of 6-logs in infectivity post-inactivation and monitoring cytopathic effect and viral mRNA over three serial 7-day passages each in Vero E6 cells. The X-ray inactivated SARS-CoV-2 was sequenced as previously described [32].

## 5.5. ELISA

For passive absorption, high binding Nunc 96-well microplates from Sigma were first rinsed with PBS (3 × 200 µl). To each well, we added 100 µl of purified protein (C1-Fc or H4) at 5 µg ml$^{-1}$ and incubated for 3 h at 37°C. The plate was then washed with PBS and blocked by the addition of 3% milk/PBS (200 µl well$^{-1}$) at 4°C overnight. After a further wash with PBS, the recombinant SARS-CoV-2 Spike protein was added in serial dilutions ranging from 100 µg ml$^{-1}$ to 0.01 µg ml$^{-1}$ and

incubated for 90 min at 37°C with PBS as a negative control. The wells were washed again with PBS and 100 µl well$^{-1}$ of HRP-conjugated probe molecules diluted (from 0.5 mg ml$^{-1}$) in PBS (1 : 3000) and added to each well and incubated at room temperature for 2 h. The wells were washed again with PBS, before a freshly prepared solution of the developer was added.

For biotinylated nanobodies, streptavidin-coated high capacity 96-well microplates from Sigma were washed with Tris–HCl 50 mM, NaCl 150 mM, 0.1% BSA and 0.05% Tween-20 (EWB). A 100 µl solution of biotinylated nanobodies previously diluted to 0.5 µg ml$^{-1}$ in ELISA wash buffer (EWB) was added to each well and incubated at room temperature for 2 h. The plate was washed with EWB and 100 µl of a serial dilution of Spike protein (typically 10–0.0001 µg ml$^{-1}$) were arrayed in the wells followed by 30 min incubation at room temperature. Following a subsequent wash step with EWB, 100 µl/HRP-Fc-conjugated nanobodies (0.5 mg ml$^{-1}$) were diluted in EWB (1 : 1000) was added to each well and left for 30 min at room temperature.

Two developers were used, firstly 100 µl ABTS (0.3 mg ml$^{-1}$) in a peroxide solution (0.01%) was added and the plate was shielded from light and left for 20 min, at which point the absorbance at 405 and 410 nm was read by SpectraMax M3 microplate reader (Molecular Devices). For the second developer, 100 µl TMB (0.2 mg ml$^{-1}$) in a peroxide solution (0.01%), after which the plate was shielded from light for 20 min. $H_2SO_4$ (2 M, 100 µl well$^{-1}$) was added to each well the OD at 450 nm was recorded using the same SpectraMax M3 microplate reader as above. All samples were run in triplicate, and the data were analysed using GraphPad Prism v. 9.

## 5.6. ELISA with virus

C5-Fc was specifically biotinylated as above and the probe nanobody chosen was F2-Fc conjugated to HRP as described above. Streptavidin-coated plates and the TMB development protocols described above were used. Three different means of inactivation of SARS-CoV-2 were employed addition of detergent (Empigen) followed heating (56°C, 30 min); addition of 4% FA and exposure to 12.2 kGy of X-ray irradiation. A pseudotyped lentivirus NL4.3 expressing SARS-CoV-2 Spike protein was also evaluated and was supplied as live virus in PBS. SARS-CoV-2 virus samples were tested in serial dilutions ranging from 4000 ffu ml$^{-1}$ to 16.384 ffu ml$^{-1}$. Pseudovirus dilution series ranged from 2500 TCID50 ml$^{-1}$ to 10.24 TCID50 ml$^{-1}$.

## 5.7. Limit of detection

The sensitivity of ELISA is largely judged by the LOD; the lowest detectable level of analyte that can reliably be distinguished from the background. We used LOD = $(3.3 * Sy)/k$, where Sy is the standard deviation of blank replicate samples, and $k$ is the gradient of slope from linear regression analysis. As standard deviation can vary significantly, $k$ can be a helpful way to gauge sensitivity.

Data accessibility. The figures in the paper present the raw data. Plasmids of C1, C5, F2, H3 and H4 are available at ADDGENE with IDs 171924, 171925, 171926, 171927 and 160313, respectively.

The data are provided in the electronic supplementary material [33].

Authors' contributions. G.C.G. and A.L. carried out all the ELISA experimental work, J.H., J.D. and C.N. supplied proteins used in the study, B.A., A.H. and W.J. supplied inactivated virus, R.J.O. and J.H.N. designed the study. All authors analysed the data.

Competing interests. J.H., J.H.N. and R.J.O. are named on patents that describe the nanobodies discussed in this study. The other authors have no competing interests.

Funding. This work was support by the Rosalind Franklin Institute funding delivery partner EPSRC, Wellcome Trust (grant no. 100209/Z/12/Z) and Biological Sciences Research Council (BBSRC; BB/T006161/1). Containment level 3 experiments were funded through the generous support of philanthropic donors to the University of Oxford's COVID-19 Research Response Fund.

Acknowledgements. We thank Rebecca Moore at Dunn School Oxford University for help in producing the virus.

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
