## [Peer Review File · Royal Society Open Science]

Review History

RSOS-211016.R0 (Original submission)

Review form: Reviewer 1 (Vijay Gupta)

Is the manuscript scientifically sound in its present form?

Yes

Are the interpretations and conclusions justified by the results?

Yes

Is the language acceptable?

Yes

Do you have any ethical concerns with this paper?

No

Have you any concerns about statistical analyses in this paper?

No

Recommendation?

Accept with minor revision (please list in comments)

Comments to the Author(s)

In this manuscript titled as "The use of nanobodies in a sensitive ELISA test for SARSCoV-2 antigens" by Dr. Naismith group, the authors have identified an ideal combination of their previously published nanobodies (Nbs) by perfecting the combination and selectively labeling them to be used in ELISA assays to detect SARS-CoV2 antigens. The manuscript is written coherently and explain all the experiments in proper details. It's a pity that authors have not declared the sequence of their Nbs due to IP issues which would have greatly benefitted other researchers. Nonetheless, the methodology described in the paper along with the findings will help in advancing the field.

The authors should atleast discuss their important finding in the light of neutralization assay and whether their combination of Nbs could also have better neutralization efficiency and used as therapeutics. I have only one minor issue that since this work is going to be utilized for the development of real world diagnostics, the authors could demonstrate the usage of their Nbs combination and methodology in few clinical samples if possible, if not its Ok.

Review form: Reviewer 2

Is the manuscript scientifically sound in its present form?

No

Are the interpretations and conclusions justified by the results?

No

Is the language acceptable?

Yes

Do you have any ethical concerns with this paper?

No

Have you any concerns about statistical analyses in this paper?

Yes

Recommendation?

Major revision is needed (please make suggestions in comments)

Comments to the Author(s)

The article submitted by Girt et al. describes the development and optimization of a sandwich ELISA based on the use of nanobodies for the determination of Spike 1 protein, relevant biomarker for the determination of SARS-CoV-2 infection. The authors showed the experiments carried out to optimize the sensitivity of the assay by exploring different methods of immobilization as well different nanobody fusion proteins or chemically modified (biotinylation by non-specific and site-specific labelling). The assay optimized resulted a sensitive assay and was applied for the measurement of Spike 1 protein, pseudo-virus and inactivated virus. In general, the article is well-structured and the conclusions are supported by the data, however

some issues should be modified and corrected prior publishing in RSOS journal. The article has a great potential and I hope that the editor and authors accept my suggestions and comments.

General comments:

- My concern is about the applicability of the developed assay. The assay has a great potential but it was not applied for the problem that the society is facing that is the detection of the virus in patients. I encourage the authors to test the assay in real matrices such as naso-faringeal or oro-faringeal samples.
- Nowadays, several SARS-CoV-2 mutants have infect worldwide. The authors have to demonstrate that the developed assay is detecting also the main current variant of concern (alpha, beta, gamma and delta, at least). Selectivity assay is required.
- I missed further analytical characterization such as reproducibility and accuracy studies.
- The authors claimed that in order to demonstrated the robustness of the system, they analysed four different batches of heat-Empigen treatment. According the data obtained, the assay is different for each batch, so the treatment is not robust or the assay is sensitive to the matrix. In my opinion, this experiement should be discussed more in detail an explain the reason of this behavior.

Specific comments

- The title should reflect that specifically the assay is developed for the detection of Spike 1 protein
- The abstract doesn't explain the main investigations carried out in the article. In my opinion, the authors should be concise in the optimization carried out and omit explanation about how it is working an ELISA.
- Page 3, line 16-18. The sentence about the causality of the COVID-19 disease have to be more specific and a reference is required.
- Page 3, line 43-45. The authors refer PCR as "the most accurate approach to detect RNA...". RT-PCR or qPCR are considered by FDA and ECDC as the golden standard for the diagnosis of COVID-19 with the highest sensitivity and specificity. I suggest to describe PCR in this way, not only considering their accuracy.
- Page 4 Line7-34. ELISA and LFIA are well-known techniques that it is no need to explain their basis. I suggest to be focused in the current methodologies available in the market for the detection of SARS-CoV-2 antigens or what it can be found in the literature for the detection of antigens. Any mention has been done in the current state of the art of the detection of SARS-CoV-2 antigens, specifically for S1 protein.
- Page Line 32. Usually the detection antibody label with the nanoparticle are immobilized in the conjugate pad from the rapid test, therefore, once the sample is contact with the LFIA, the same sample reconstitutes the detection antibody and together flow through the membrane to reach the test line. It is true that the proper sample is acting as "mobile phase", but that's it.
- Figure 1. According the text, the authors tries to explain that the nanobodies chosen recognize differnet epitopes. In the figure legend says "In yellow space fill is a nanobody that recognises the ACE2 epitope....", but the figure represents the nanobody that recognize the RBD region. Please, clarify it.
- Page 6, line 20-24. I agree that one explanation is the VHH difficulties to be oriented for the biorecognition. FYI, Li et al. has concluded in the same direction (<https://pubs.acs.org/doi/10.1021/acs.analchem.0c01115>)
- Page 6 Line 36. I would suggest omitting "ThermoFisher EZ-Link Sulfo-NHS-Biotin" reagent. It is a brand name that is specified in the materials and methods section.
- Figure 2 and others. It is not explained what represent the dotted line at 1.5.
- Page 7 Line 37. I would recommend to use concentration despite dilution factor taking into consideration that the authors known it.
- Page 7 Line 42-49. The authors showed that H4-Fc-HRP conjugate is not useful, but the previous H4-HRP was functional. Which is the reason?
- Page 7 Line 55 and Table 1. The authors are refereeing to the regression gradient but it is not explained how it is calculated and the meaning of this value.

- Figure 3. In my opinion, this figure is meaningful
- Figure 4 and 5, and other related in the SI file. I think that would be enough to show the linear regression graph because a non-linear regression fitting is not used to extract LOD or other analytical parameters.
- Table 2 and 3. What is St. dev.? I think that R2 is enough to determine the linearity of the analysed points. Also, I miss the o.o. from the linear regression fitting in the table. Moreover, the title says about estimation of LOD. I would prefer to say “calculated”.
- Figure 5. I would avoid sentence like “Using C5-Fc-SS-biotin and F2-Fc-HRP improved detection sensitivity” , it is not an assessment. The authors should be more focused on the description of the figure and how it was performed the assay. This fact is also applicable to Figure 4.
- I recommend homogenizing units. In table 1 is described in ng/mL but in the Discussion section is expressed in pg/mL. Also, in Page 7 Line 41, S1 protein concentration is expressed in μM , but in the abstract, the authors said that reach sub-picomolar.
- Page 17 Line 42. The authors used a kit for the quantification of the biotinylation process, but any mention has been done in the article describing how many biotins have been attached.
- Page 20 ELISA section. The authors have to describe how many replicates have used for the generation of each point in the constructed calibration curves.
- Page 21 Line 41. The authors have omitted in the formula the subtraction of the o.o.
- Page 23. Ref 14. It is not acceptable to reference an article that it is not already accepted.
- References section. Homogenize the format, especially in articles where the doi is not described.

According my revision, in my opinion this article is not suitable for publication in RSOS in the present form.

Review form: Reviewer 3

Is the manuscript scientifically sound in its present form?

Yes

Are the interpretations and conclusions justified by the results?

Yes

Is the language acceptable?

Yes

Do you have any ethical concerns with this paper?

No

Have you any concerns about statistical analyses in this paper?

No

Recommendation?

Accept with minor revision (please list in comments)

Comments to the Author(s)

This paper describes the use of nanobodies in the development of sensitive ELISA assays of the Spike protein of SARS-CoV-2.

In this study, the authors explore a number of nanobodies discovered in a library screen in two earlier studies, one of which is under review with another journal. Their goal here is to evaluate

the suitability of these nanobodies as capture and detection agents in an ELISA assay. The variables are: the choice of capture nanobody; the choice of detection nanobody; and the use of biotin and streptavidin as a capture system. The paper explores recognition of the Spike protein itself and its recognition when presented on a viral surface.

The work has been performed well. It is not hypothesis-led nor is it especially novel, rather it documents the development of an assay. It is very topical, concerning as it does, detection of an antigen from a virus which is causing a pandemic.

The context of the work is well set out in the Introduction and Discussion sections. The Results section could be improved by giving more background on the reagents used (see Points) for those not fully familiar with immunodetection assays.

Points

In the final paragraph on page 4, the virtues of the small size of nanobodies are mentioned. Perhaps then explain why these are later conjugated to Fc molecules. How is the Fc-conjugation achieved?

Figure 2. Title is misleading as (b) includes absorption using biotin which is not passive?

On page 7, remind reader of origin of C1, H4 etc, it is mentioned in the Introduction but the terms are not memorable and a reminder will help. Define Nb as abbreviation for nanobody.

On page 9, Some brief explanation/ definition of the following would be useful

- Pseudotyped NL4.3 HIV-1 backbone virus
- TCID50
- Empigen
- ffu

The description of the site specific biotinylation is not clear. What is the specific site of biotinylation and how was this achieved? This is important to know and should not have to be deduced from a Supplementary Figure Legend.

Decision letter (RSOS-211016.R0)

Dear Professor Naismith

On behalf of the Editors, we are pleased to inform you that your Manuscript RSOS-211016 "The use of nanobodies in a sensitive ELISA test for SARS-CoV-2 antigens" has been accepted for publication in Royal Society Open Science subject to minor revision in accordance with the referees' reports. Please find the referees' comments along with any feedback from the Editors below my signature.

Please submit your revised manuscript and required files (see below) no later than 7 days from today's (ie 31-Aug-2021) date. Note: the ScholarOne system will 'lock' if submission of the revision is attempted 7 or more days after the deadline. If you do not think you will be able to meet this deadline please contact the editorial office immediately.

on behalf of Professor Jo Parish (Associate Editor) and Malcolm White (Subject Editor)
openscience@royalsociety.org

Reviewer comments to Author:

Reviewer: 1

Comments to the Author(s)

In this manuscript titled as "The use of nanobodies in a sensitive ELISA test for SARSCoV-2 antigens" by Dr. Naismith group, the authors have identified an ideal combination of their previously published nanobodies (Nbs) by perfecting the combination and selectively labeling them to be used in ELISA assays to detect SARS-CoV2 antigens. The manuscript is written coherently and explains all the experiments in proper details. It's a pity that authors have not declared the sequence of their Nbs due to IP issues which would have greatly benefitted other researchers. Nonetheless, the methodology described in the paper along with the findings will help in advancing the field.

The authors should atleast discuss their important finding in the light of neutralization assay and whether their combination of Nbs could also have better neutralization efficiency and used as therapeutics. I have only one minor issue that since this work is going to be utilized for the development of real world diagnostics, the authors could demonstrate the usage of their Nbs combination and methodology in a few clinical samples if possible, if not its Ok.

Reviewer: 2

Comments to the Author(s)

The article submitted by Girt et al. describes the development and optimization of a sandwich ELISA based on the use of nanobodies for the determination of Spike 1 protein, relevant biomarker for the determination of SARS-CoV-2 infection. The authors showed the experiments carried out to optimize the sensitivity of the assay by exploring different methods of immobilization as well different nanobody fusion proteins or chemically modified (biotinylation by non-specific and site-specific labelling). The assay optimized resulted a sensitive assay and was applied for the measurement of Spike 1 protein, pseudo-virus and inactivated virus. In general, the article is well-structured and the conclusions are supported by the data, however

some issues should be modified and corrected prior publishing in RSOS journal. The article has a great potential and I hope that the editor and authors accept my suggestions and comments.

General comments:

- My concern is about the applicability of the developed assay. The assay has a great potential but it was not applied for the problem that the society is facing that is the detection of the virus in patients. I encourage the authors to test the assay in real matrices such as naso-faringeal or oro-faringeal samples.
- Nowadays, several SARS-CoV-2 mutants are extant worldwide. The authors have to demonstrate that the developed assay is detecting also the main current variants of concern (alpha, beta, gamma and delta, at least), or at least discuss this in more detail.
- I missed further analytical characterization such as reproducibility and accuracy studies.
- The authors claimed that in order to demonstrate the robustness of the system, they analysed four different batches of heat-Empigen treatment. According the data obtained, the assay is different for each batch, so the treatment is not robust or the assay is sensitive to the matrix. In my opinion, this experiement should be discussed more in detail an explain the reason of this behavior.

Specific comments

- The title should reflect that specifically the assay is developed for the detection of Spike 1 protein
- Page 3, line 16-18. The sentence about the causality of the COVID-19 disease has to be more specific and a reference is required.
- Page 3, line 43-45. The authors refer PCR as “the most accurate approach to detect RNA...”. RT-PCR or qPCR are considered by FDA and ECDC as the golden standard for the diagnosis of COVID-19 with the highest sensitivity and specificity. I suggest to describe PCR in this way, not only considering their accuracy.
- Page 4 Line7-34. ELISA and LFIA are well-known techniques that it is no need to explain their basis. I suggest to be focused in the current methodologies available in the market for the detection of SARS-CoV-2 antigens or what it can be found in the literature for the detection of antigens. Any mention has been done in the current state of the art of the detection of SARS-CoV-2 antigens, specifically for S1 protein.
- Page Line 32. Usually the detection antibody label with the nanoparticle are immobilized in the conjugate pad from the rapid test, therefore, once the sample is contact with the LFIA, the same sample reconstitutes the detection antibody and together flow through the membrane to reach the test line. It is true that the proper sample is acting as “mobile phase”, but that’s it.
- Figure 1. According the text, the authors tries to explain that the nanobodies chosen recognize differnet epitopes. In the figure legend says “In yellow space fill is a nanobody that recognises the ACE2 epitope....”, but the figure represents the nanobody that recognize the RBD region. Please, clarify it.
- Page 6, line 20-24. I agree that one explanation is the VHH difficulties to be oriented for the biorecognition. FYI, Li et al. has concluded in the same direction (<https://pubs.acs.org/doi/10.1021/acs.analchem.0c01115>)
- Page 6 Line 36. I would suggest omitting “ThermoFisher EZ-Link Sulfo-NHS-Biotin” reagent. It is a brand name that is specified in the materials and methods section.
- Figure 2 and others. It is not explained what represent the dotted line at 1.5.
- Page 7 Line 37. I would recommend to use concentration despite dilution factor taking into consideration that the authors know it.
- Page 7 Line 42-49. The authors showed that H4-Fc-HRP conjugate is not useful, but the previous H4-HRP was functional. Which is the reason?
- Page 7 Line 55 and Table 1. The authors are refereeing to the regression gradient but it is not explained how it is calculated and the meaning of this value.

- Table 2 and 3. What is St. dev.? I think that R2 is enough to determine the linearity of the analysed points. Also, I miss the o.o. from the linear regression fitting in the table. Moreover, the title says about estimation of LOD. I would prefer to say “calculated”.
- Figure 5. I would avoid sentence like “Using C5-Fc-SS-biotin and F2-Fc-HRP improved detection sensitivity” , it is not an assessment. The authors should be more focused on the description of the figure and how the assay was performed. This fact is also applicable to Figure 4.
- I recommend homogenizing units. In table 1 is is described in ng/mL but in the Discussion section is expressed in pg/mL. Also, in Page 7 Line 41, S1 protein concentration is expressed in μM , but in the abstract, the authors said that reach sub-picomolar.
- Page 17 Line 42. The authors used a kit for the quantification of the biotinylation process, but any mention has been done in the article describing how many biotins have been attached.
- Page 20 ELISA section. The authors have to describe how many replicates have used for the generation of each point in the constructed calibration curves.
- Page 21 Line 41. The authors have omitted in the formula the subtraction of the o.o.
- Page 23. Ref 14. Please check whether this has now been published and update.
- References section. Homogenize the format, especially in articles where the doi is not described.

Reviewer: 3

Comments to the Author(s)

This paper describes the use of nanobodies in the development of sensitive ELISA assays of the Spike protein of SARS-CoV-2.

In this study, the authors explore a number of nanobodies discovered in a library screen in two earlier studies, one of which is under review with another journal. Their goal here is to evaluate the suitability of these nanobodies as capture and detection agents in an ELISA assay. The variables are: the choice of capture nanobody; the choice of detection nanobody; and the use of biotin and streptavidin as a capture system. The paper explores recognition of the Spike protein itself and its recognition when presented on a viral surface.

The work has been performed well. It is not hypothesis-led nor is it especially novel, rather it documents the development of an assay. It is very topical, concerning as it does, detection of an antigen from a virus which is causing a pandemic.

The context of the work is well set out in the Introduction and Discussion sections. The Results section could be improved by giving more background on the reagents used (see Points) for those not fully familiar with immunodetection assays.

Points

In the final paragraph on page 4, the virtues of the small size of nanobodies are mentioned. Perhaps then explain why these are later conjugated to Fc molecules. How is the Fc-conjugation achieved?

Figure 2. Title is misleading as (b) includes absorption using biotin which is not passive?

On page 7, remind reader of origin of C1, H4 etc, it is mentioned in the Introduction but the terms are not memorable and a reminder will help. Define Nb as abbreviation for nanobody.

On page 9, Some brief explanation/ definition of the following would be useful

- Pseudotyped NL4.3 HIV-1 backbone virus
- TCID50
- Empigen

- ffu

The description of the site specific biotinylation is not clear. What is the specific site of biotinylation and how was this achieved? This is important to know and should not have to be deduced from a Supplementary Figure Legend.

===PREPARING YOUR MANUSCRIPT===

===PREPARING YOUR REVISION IN SCHOLARONE===

Author's Response to Decision Letter for (RSOS-211016.R0)

See Appendix A.

Decision letter (RSOS-211016.R1)

Dear Professor Naismith,

I am pleased to inform you that your manuscript entitled "The use of nanobodies in a sensitive ELISA test for SARS-CoV-2 Spike 1 protein" is now accepted for publication in Royal Society Open Science.

COVID-19 rapid publication process:

We are taking steps to expedite the publication of research relevant to the pandemic. If you wish, you can opt to have your paper published as soon as it is ready, rather than waiting for it to be published the scheduled Wednesday.

This means your paper will not be included in the weekly media round-up which the Society sends to journalists ahead of publication. However, it will still appear in the COVID-19 Publishing Collection which journalists will be directed to each week (<https://royalsocietypublishing.org/topic/special-collections/novel-coronavirus-outbreak>).

If you wish to have your paper considered for immediate publication, or to discuss further, please notify openscience_proofs@royalsociety.org and press@royalsociety.org when you respond to this email.

on behalf of Professor Jo Parish (Associate Editor) and Malcolm White (Subject Editor)
openscience@royalsociety.org

Appendix A

We thank the reviewers for their very helpful comments. Our responses are in **bold red**.

Reviewer comments to Author:

Reviewer: 1

Comments to the Author

In this manuscript titled as “The use of nanobodies in a sensitive ELISA test for SARSCoV-2 antigens” by Dr. Naismith group, the authors have identified an ideal combination of their previously published nanobodies (Nbs) by perfecting the combination and selectively labeling them to be used in ELISA assays to detect SARS-CoV2 antigens. The manuscript is written coherently and explains all the experiments in proper details. It’s a pity that authors have not declared the sequence of their Nbs due to IP issues which would have greatly benefitted other researchers. Nonetheless, the methodology described in the paper along with the findings will help in advancing the field

Our apologies, the nanobody sequences were already available in reference 13, 14

Reference 14 was online <https://www.researchsquare.com/article/rs-548968/v1> was a pre-print

We had no intention of not disclosing, rather we had already disclosed but we had not made this clear, our apologies.

Statement now added

“Purified H4, H4-Fc, C5-Fc, C1-Fc, F2-Fc, RBD and SARS-CoV-2 Spike were prepared as previously described(13,14); these reports also contain the nanobodies sequences”

The authors should at least discuss their important finding in the light of neutralization assay and whether their combination of Nbs could also have better neutralization efficiency and used as therapeutics.

Reference 14 reports the therapeutic effect of C5, we have now added

“We have shown that C5 has a potent therapeutic effect when administered topically or intraperitoneally to hamsters(14). This combined with our previous observation of additive neutralisation between non-overlapping neutralising agents(13) suggests that C1 or F2 could be effective in combination with C5.”

I have only one minor issue that since this work is going to be utilized for the development of real world diagnostics, the authors could demonstrate the usage of their Nbs combination and methodology in a few clinical samples if possible, if not its Ok.

The concept behind the assay was originally to measure Spike as a biotechnology tool. We were and remain unable to access clinical samples in a suitably equipped CL3 environment to carry out this assay on.

Reviewer: 2

Comments to the Author(s)

The article submitted by Girt et al. describes the development and optimization of a sandwich ELISA based on the use of nanobodies for the determination of Spike 1 protein, relevant biomarker for the determination of SARS-CoV-2 infection. The authors showed the experiments carried out to optimize the sensitivity of the assay by exploring different methods of immobilization as well different nanobody fusion proteins or chemically modified (biotinylation by non-specific and site-specific labelling). The assay optimized resulted a sensitive assay and was applied for the measurement of Spike 1 protein, pseudo-virus and inactivated virus. In general, the article is well-structured and the conclusions are supported by the data, however some issues should be modified and corrected prior publishing in RSOS journal. The article has a great potential and I hope that the editor and authors accept my suggestions and comments.

General comments:

My concern is about the applicability of the developed assay. The assay has a great potential but it was not applied for the problem that the society is facing that is the detection of the virus in patients. I encourage the authors to test the assay in real matrices such as naso-faringeal or oro-faringeal samples.

We concur with the reviewer and reviewer 1, but we were and are unable to access clinical samples in a suitably equipped CL3 environment to carry out this assay. Our hope is that having published others will take the materials into this setting.

Nowadays, several SARS-CoV-2 mutants are extant worldwide. The authors have to demonstrate that the developed assay is detecting also the main current variants of concern (alpha, beta, gamma and delta, at least), or at least discuss this in more detail.

This is a very good suggestion. Paper 14 discusses the nanobody binding to these variants. In short, E484 mutations oblates C5 binding. C1 and F2 remain cross reactive.

We have performed the RBD binding for beta and delta (Figure s7). As we expected delta is detected less well than Victoria and beta is not detected at all.

I missed further analytical characterization such as reproducibility and accuracy studies.

The referee raises a useful point. With the purified proteins, the results are consistent between batches. However, (below) with the virus the results are more variable. We do not have the ability to analytically determine this uncertainty; in part because our suspicion is that the dominant factor is variability between viral preparations, but this is very hard to capture and quantitate.

The authors claimed that in order to demonstrate the robustness of the system, they analysed four different batches of heat-Empigen treatment. According the data obtained, the assay is different for each batch, so the treatment is not robust or the assay is sensitive to the matrix. In my opinion, this experiment should be discussed more in detail and explain the reason of this behavior.

To continue from above, the purified proteins behave consistently but the behaviour with virus varies. We say

“Since purified RBD and Spike proteins give highly consistent results, we attribute the observed assay variability to viral batches.”

I do not wish to split hairs, the referee is quite right about the variability (the larger point). We did use the phrase

“In order to probe the robustness of the assay,”

The word “probe” not “demonstrate” was chosen.

Specific comments

The title should reflect that specifically the assay is developed for the detection of Spike 1 protein
Changed to “SARS-CoV-2 Spike 1 protein”

Page 3, line 16-18. The sentence about the causality of the COVID-19 disease has to be more specific and a reference is required.

We did state,

“Severe acute respiratory syndrome coronavirus 2 (SARS-CoV-2) is the novel human coronavirus (HCoV) responsible for the COVID-19 pandemic”

We are unclear what the referee wants at line 16, sorry.

Page 3, line 43-45. The authors refer PCR as “the most accurate approach to detect RNA...”. RT-PCR or qPCR are considered by FDA and ECDC as the golden standard for the diagnosis of COVID-19 with the highest sensitivity and specificity. I suggest to describe PCR in this way, not only considering their accuracy.

Changed as suggested

Page 4 Line7-34. ELISA and LFIA are well-known techniques that it is no need to explain their basis. I suggest to be focused in the current methodologies available in the market for the detection of SARS-CoV-2 antigens or what it can be found in the literature for the detection of antigens. Any mention has been done in the current state of the art of the detection of SARS-CoV-2 antigens, specifically for S1 protein.

Changed as suggested

Page Line 32. Usually the detection antibody label with the nanoparticle are immobilized in the conjugate pad from the rapid test, therefore, once the sample is contact with the LFIA, the same sample reconstitutes the detection antibody and together flow through the membrane to reach the test line. It is true that the proper sample is acting as “mobile phase”, but that’s it.

Changed to:

“the capture antibodies are often immobilised on nitrocellulose and the probe antibody, modified with a colorimetric nanoparticle, is incubated with the test sample, thus becoming the “mobile phase” of the device.”

Figure 1. According the text, the authors tries to explain that the nanobodies chosen recognize different epitopes. In the figure legend says “In yellow space fill is a nanobody that recognises the

ACE2 epitope....”, but the figure represents the nanobody that recognize the RBD region. Please, clarify it.

Caption clarified, changed to:

“Figure 1 The two non-overlapping epitopes used for the ELISA. In yellow space fill is a nanobody that recognises the ACE2 RBD epitope(13, 14) known as cluster 2 (15) (group 1(18)) epitope and in blue a second nanobody(14) which recognises a different RBD epitope, first described in the CR3022 structure(16, 17), known as cluster 1 (15) (group 4(18)) epitope.”

Page 6, line 20-24. I agree that one explanation is the VHH difficulties to be oriented for the biorecognition. FYI, Li et al. has concluded in the same direction (<https://pubs.acs.org/doi/10.1021/acs.analchem.0c01115>)

We apologise for the poor clarity we had cited this reference in the discussion but not in the results (as the referee says).

“Consistent with other reports using nanobodies in sandwich ELISA(25), the direct adsorption of nanobodies onto simple plates gave an ELISA that was less successful than biotinylated protein and streptavidin coated plates.”

To clarify, we have removed the sentence about why VHH are not as good from results and more fully dealt with the issue in the discussion. We have replaced the sentence above with

“Direct absorbance of the V_HH domain of the nanobody onto plates was not successful in an ELISA against the Spike protein. We concluded either VHH was not readily adsorbed onto the plate or when absorbed onto the plate its binding site was obscured. Other workers (25) targeting a different protein have observed similar issues with V_HH domains and drawn similar conclusions. Switching to biotinylated protein and streptavidin coated plates gave much better results.”

Page 6 Line 36. I would suggest omitting “ThermoFisher EZ-Link Sulfo-NHS-Biotin” reagent. It is a brand name that is specified in the materials and methods section.

Changed to: prepared with a Sulfo-NHS-Biotin reagent.

Figure 2 and others. It is not explained what represent the dotted line at 1.5.

“Absorbance values above the dotted line were not actually measured, rather the sample was first diluted then measured.”

• Page 7 Line 37. I would recommend to use concentration despite dilution factor taking into consideration that the authors know it.

Changed to 0.5 µg/mL

• Page 7 Line 42-49. The authors showed that H4-Fc-HRP conjugate is not useful, but the previous H4-HRP was functional. Which is the reason?

We do not know. We know H4-Fc sticks to other proteins, we did not test H4-HRP because we moved on from the V_HH on their own.

Page 7 Line 55 and Table 1. The authors are refereeing to the regression gradient but it is not explained how it is calculated and the meaning of this value.

K has no meaning, rather as the gradient derived from regression analysis, it informs about sensitivity. A large k, means small changes in analyte give large responses. It is not an absolute value but is helpful when comparing similar experiments.

Table 2 and 3. What is St. dev.? I think that R2 is enough to determine the linearity of the analysed points. Also, I miss the o.o from the linear regression fitting in the table. Moreover, the title says about estimation of LOD. I would prefer to say “calculated”.

Explanation of st dev now added.

Calculation of limit of detection using slope and standard deviation is a standard method. R-squared is included to provide evidence for the robustness of the data points.

Table captions are changed to “calculated”.

O.O (optical zero) is an not explicit factor in this method for LOD calculation.

Figure 5. I would avoid sentence like “Using C5-Fc-SS-biotin and F2-Fc-HRP improved detection sensitivity” , it is not an assessment. The authors should be more focused on the description of the figure and how the assay was performed. This fact is also applicable to Figure 4.

A very good point

Figure 4 changed to: ELISA using biotin_x-C5-Fc and F2-Fc-HRP combination we were able to measure antigen when presented in viral form.

Figure 5 changed to: ELISA using C5-Fc-SS-biotin and F2-Fc-HRP against two recombinant antigens and two viral samples

I recommend homogenizing units. In table 1 is is described in ng/mL but in the Discussion section is expressed in pg/mL. Also, in Page 7 Line 41, S1 protein concentration is expressed in μM , but in the abstract, the authors said that reach sub-picomolar.

All now in pg, this is a good suggestion and improves the flow.

In P7 Line 41, Concentration was written as μM rather than $\mu\text{g/mL}$ – this has now been corrected.

Page 17 Line 42. The authors used a kit for the quantification of the biotinylation process, but any mention has been done in the article describing how many biotins have been attached.

typically 4-6 biotin moieties for Fc-conjugated nanobodies, and 1-3 biotin moieties for VHH domain nanobodies. Now added

Page 20 ELISA section. The authors have to describe how many replicates have used for the generation of each point in the constructed calibration curves.

Added: All samples were run in triplicate, and the data were analyzed using GraphPadPrism v9.

Page 21 Line 41. The authors have omitted in the formula the subtraction of the o.o.

As above.

Page 23. Ref 14. Please check whether this has now been published and update.

- References section. Homogenize the format, especially in articles where the doi is not described.

DONE

Reviewer: 3

Comments to the Author(s)

This paper describes the use of nanobodies in the development of sensitive ELISA assays of the Spike protein of SARS-CoV-2.

In this study, the authors explore a number of nanobodies discovered in a library screen in two earlier studies, one of which is under review with another journal. Their goal here is to evaluate the suitability of these nanobodies as capture and detection agents in an ELISA assay. The variables are: the choice of capture nanobody; the choice of detection nanobody; and the use of biotin and streptavidin as a capture system. The paper explores recognition of the Spike protein itself and its recognition when presented on a viral surface.

The work has been performed well. It is not hypothesis-led nor is it especially novel, rather it documents the development of an assay. It is very topical, concerning as it does, detection of an antigen from a virus which is causing a pandemic.

The context of the work is well set out in the Introduction and Discussion sections. The Results section could be improved by giving more background on the reagents used (see Points) for those not fully familiar with immunodetection assays.

Points

In the final paragraph on page 4, the virtues of the small size of nanobodies are mentioned. Perhaps then explain why these are later conjugated to Fc molecules. How is the Fc-conjugation achieved

Added:

“To account for limitations of nanobodies in ELISA, such as poor adhesion to the plate via passive adsorption and obstruction of the binding motif due to their small size, we prepared V_{HH} conjugates with human IgG1 Fc (bivalent and glycosylated). This not only increased the overall size of the nanobodies (whilst still keeping them significantly smaller than antibodies), but also introduced regions that could be site selectively modified distal to the V_{HH} domain, encouraging adhesion to the plate in optimised orientation, distal from the binding motif.”

Figure 2. Title is misleading as (b) includes absorption using biotin which is not passive?

Changed to:

“Figure 2 Immobilising the capture agent by passive absorption vs biotinylation.”

On page 7, remind reader of origin of C1, H4 etc, it is mentioned in the Introduction but the terms are not memorable and a reminder will help. Define Nb as abbreviation for nanobody.

Added: “cluster 1 ACE2 epitope biner” / “cluster CR3022 epitope binder” (where Nbs are first mentioned in results)

Added: “We introduced a further two nanobodies: F2 (cluster 1 ACE2 epitope binder) and C5 (cluster 2 CR3022 epitope binder)”

Added: “(Nbs)” to first mention of nanobodies in main text.

On page 9, Some brief explanation/definition of the following would be useful

Pseudotyped NL4.3 HIV-1 backbone virus

Added:

“This pseudovirus is based on the genomic backbone of HIV-1, with two frameshifts which render it Env (viral envelope protein) negative and Vpr (viral protein R) negative. Without the expression of these two proteins, this clone is competent for a single round of replication and non-infectious”

TCID50 Empigen

Added: Viral titres were quantified by either tissue culture infectious dose 50 (TCID50) assay, or focal forming assay (FFA).

ffu

Added: c) Empigen (a zwitterionic detergent) (0.05%) and heat (60 °C, 30 minutes).

The description of the site specific biotinylation is not clear. What is the specific site of biotinylation and how was this achieved? This is important to know and should not have to be deduced from a Supplementary Figure Legend.

Now added

“Site specific functionalization of antibodies by tagging native antibody residues has been achieved using microbial transglutaminase (mTGase)(21,22). mTGase recognizes and tags the glutamine residue in the conserved sequence ‘PREEQYNXT’ in the Fc region of antibodies. Previous studies have used this method for site specific conjugation of radioactive probes, fluorescent dyes as well as introducing orthogonal functional tags in the Fc region of antibodies(23). Since the capture nanobody, namely C5Fc, used in our ELISA is fused with an Fc tail and hence has the conserved ‘tag’ sequences, we decided to employ enzymatic transglutamination to specifically functionalize the glutamine residue of the PREEQYNST sequence in the Fc region (Figure 5a). In order to better expose the target glutamine residue for reaction, we cleaved the N-linked glycans using PNGase F digestion. Amine-PEG3-Biotin was added to the PNGase digested antibodies along with transglutaminase enzyme and incubated at 37 oC. The progress of the reaction was monitored by mass spectrometry (Figure 5b and S6). In addition to the required product, we consistently observed another product in our reactions which showed a loss of mass of 15 Da. We believe that addition of an internal lysine residue was a competing reaction which resulted in the observed product with a loss of mass when compared to the deglycosylated C5-Fc.”

Also Figure 5